# EnchantedBrush: Animating in Mixed Reality for Storytelling and Communication

Eve Mingxiao Li*
University of Toronto

Anran Qi†
The University of Tokyo

Mauricio Sousa‡
University of Toronto

Tovi Grossman§
University of Toronto

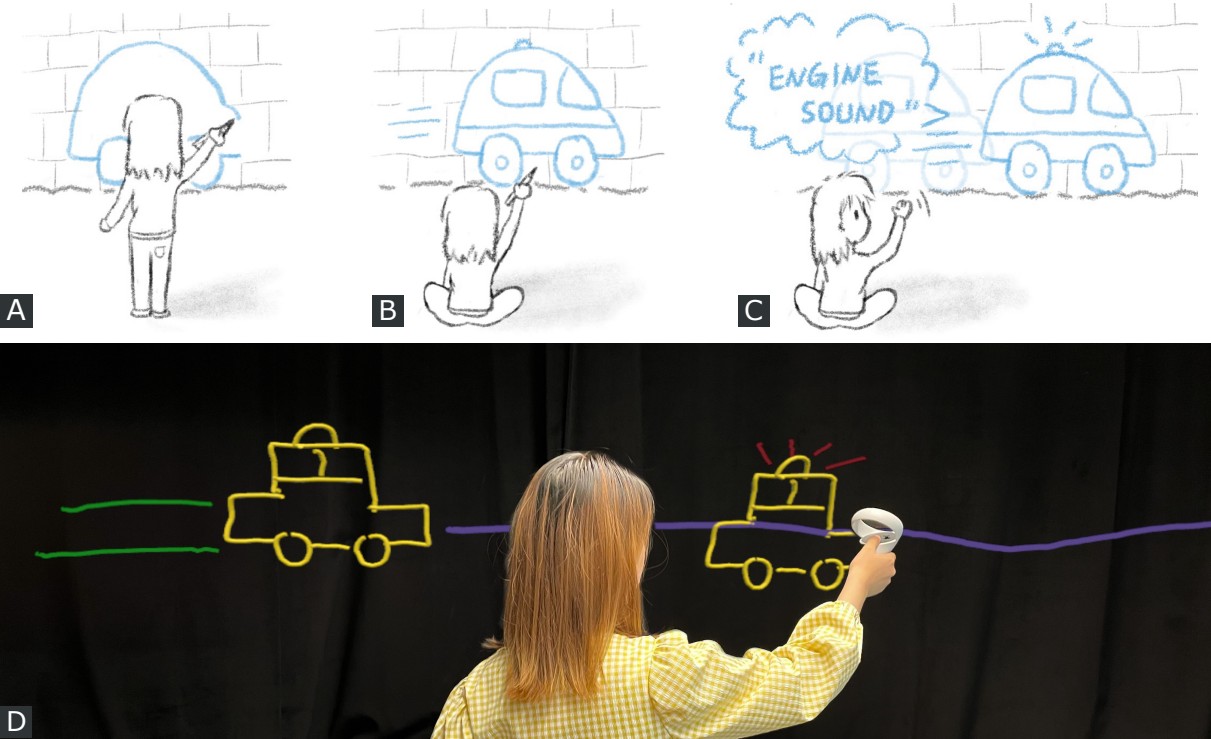

Figure 1: EnchantedBrush is a mixed-reality sketching interface for creating animated storyboards that can interact with the physical surroundings. It supports easy storytelling as if using a magical paintbrush to A) draw a vehicle and B) give it motion lines so that C) the vehicle comes to life with *automatic* motion and sound effects (the black represents elements in the real world, and the blue represents elements in the virtual world). This animation can be achieved using D) a set of interactive motion and sound brushes designed in a mixed-initiative interaction paradigm.

## ABSTRACT

Recent progress in head-worn 3D displays has made mixed-reality storytelling, which allows digital art to interact with the physical surroundings, a new and promising medium to visualize ideas and bring sketches to life. While previous works have introduced dynamic sketching and animation in 3D spaces, visual and audio effects typically need to be manually specified. We present EnchantedBrush, a novel mixed-reality approach for creating animated storyboards with *automatic* motion and sound effects in real-world environments. People can create animations that interact with their physical surroundings using a set of interactive motion and sound brushes. We evaluated our approach with 12 participants, including professional

*e-mail: eveli@dgp.toronto.edu
†e-mail: annranqi1024@g.ecc.u-tokyo.ac.jp
‡e-mail: mauricio@dgp.toronto.edu
§e-mail: tovi@dgp.toronto.edu

artists. The results suggest that EnchantedBrush facilitates storytelling and communication, and utilizing the physical environment eases animation authoring and simplifies story creation.

**Index Terms:** Human-centered computing—Human computer interaction (HCI)—Interaction paradigms—Mixed / augmented reality; Human-centered computing—Interaction design—Interaction design theory, concepts and paradigms

## 1 INTRODUCTION

Sketching, a ubiquitous communication medium for visual thinking, performs a critical role in storytelling, idea expression, and creativity [26, 29, 36, 38, 44, 45]. For more accessible animation authoring and expressive power, human-computer interaction researchers have been exploring dynamic sketching and interactive animation over decades in broad-ranging fields such as arts and storyboards [20–22], prototyping [14, 34], data visualization [28, 46], and education [40, 50]. However, these works lack the consideration of interacting with the physical surroundings.

Mixed Reality (MR) technologies further introduced new possibilities into dynamic sketching and interactive animation by expanding

the domain of interactions. The painting canvas is no longer constrained to an electronic screen but a broader canvas in the form of a freeform reality environment. This free manner has inspired numerous Virtual and Augmented Reality (VR/AR) works and commercial products with sketch-based interaction [2, 8, 19, 25, 27, 31, 35, 42]. However, these works either lack the ability to support animation and sound effects or require manual efforts to specify such effects, which could be time-consuming and redundant when creating a story. Besides, previous works also do not exploit the interaction with physical surroundings as part of sketching or storytelling. This limits the use of physical objects in communication.

In this work, we present EnchantedBrush, a novel MR interface for dynamic sketching and interactive storyboards (Figure 1). Our approach utilizes a mixed-initiative interaction paradigm — the automatic mode and the customized mode — to assist users in designing visual and sound effects. Under the automatic mode, sketched elements are animated automatically with lively sounds based on their semantic nature. Under the customized mode, users can manually define animation and add sounds. This lets users quickly create visualizations and storyboards without spending intensive time on manual effect specifications. Under both modes, virtual sketched elements can interact with the physical surroundings and generate proper sound effects. This saves users' time by releasing them from preparing object models and setting up backgrounds.

Our approach works as follows: 1) *element sketching*: users design their storyboards based on the physical surroundings and then sketch elements in mid-air. 2) *sketch recognition*: our system adopts a sketch recognition model to recognize the sketched elements and provides the top two recognition results for users to choose from. 3) *motion and sound* designing: our system will automatically add the motion and sound effects to the sketched elements according to their semantic nature. In addition to the automatic behavior, users could also customize the animation by defining motion paths and sound effects by themselves. 4) *animation*: users animate the sketched storyboard and make it interactive with the real-world environment.

We evaluated our approach with 12 participants, including three professional artists. The results show the usability of EnchantedBrush in easy storyboard creation and effective communication. We also discuss potential application scenarios based on the interviews with the participants.

To summarize, our contributions are:

- EnchantedBrush, a sketch-based interface in mixed reality for creating animated storyboards that can interact with the physical surroundings

- A mixed-initiative interaction paradigm for visual and audio design, including the automatic mode and the customized mode

- Automatic visual and audio effects for sketched objects by utilizing the semantic properties of the objects, which reduces the effort of manual specification

- User evaluation of EnchantedBrush and a set of its potential application scenarios such as for the animation industry and children teaching

## 2 RELATED WORK

Our work relates to prior research in 2D dynamic sketching interfaces, 3D animation authoring tools, and sketch-based interactions in VR/AR.

### 2.1 Dynamic Sketching Interfaces in 2D

Traditional 2D sketching applications have been well investigated and developed. Various dynamic sketching tools cover different features for animation authoring. K-Sketch [7] introduces a pen-based system to create simple prototypes of animations. Kazi et al. build

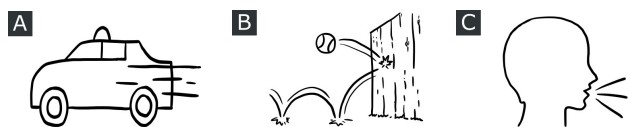

Figure 2: Motion and sound examples in storyboards, illustrations, and icons: (A) The motion lines behind the car indicate the car is running. (B) The path lines of the ball illustrate its movement trajectory. (C) The sound lines for the person show they are speaking.

Draco [21] and Kitty [20], adding kinetic textures to a collection of objects for continuous animation effects and allowing users to customize different functional relationships between object entities for interactive illustrations. Energy-Brushes [48] investigates the role of flow particles in stylized animations to create fundamental dynamics. Motion Amplifiers [22] turns the principles of 2D animation into a set of amplifiers that users can apply to animate illustrations. Sketching can also allow for natural interactions during whiteboard presentations and visualization as demonstrated by SketchStory [28], and in fluid systems such as the illustration project by Zhu et al. [50]. Moreover, Motion Doodles [43] shows how sketching can animate the motion of characters, while Vignette [23] focuses on texture creation using interactive sketch gestures. These 2D works and interfaces allow people to utilize the power and nature of sketching to create animation and texture. They create the fundamental applications of sketching. Our work keeps the core design idea of a sketching tool but expands it even further and implements the system in mixed reality to allow sketches to interact with physical environments. Besides, our system incorporates automatic animation and sounds to author lively stories, which are missing in the existing works.

### 2.2 Animation Authoring Tools in 3D

Animation tools in 3D add one more dimension to the possibility and contribute to more expressive and lively graphics. For instance, we can use human bodies to animate static 3D meshes and craft interactions with graphical elements [6,39,49]. 3D puppetry [15] provides a real-time tool for 3D model animation using physical objects as input. As for character motion, Gambaretto and Piña [9] investigate facial expressions as a way to author character animation, while Glauser et al. [12] explore a tangible interface by fluid manipulation. Guay et al. [13] study a stroke method using a space-time curve to match 3D character motion based on 2D lines. Recently, Ma et al. [32] propose stylized 3D animations in a layered authoring interface. However, these applications are still traditional screen-based interfaces. HCI researchers have further studied the potential of the interactive experience in VR. Hwang et al. [16] propose a performance-based animation system for virtual object manipulations. Another example is MagicalHands [4] which achieves animation authoring in VR using mid-air hand gestures. While these projects take the generation of animation to a new level, they make use of the interaction of body motions instead of sketching interactions. We focus on how 3D sketching in mixed reality can enhance users' ability for storytelling and idea visualization.

### 2.3 Sketch-based User Interfaces in VR and AR

Sketching in 3D is becoming an increasingly popular medium not only in pure screen-based interaction but also in MR applications. There are a number of commercial sketching and animation products in VR/AR such as Tilt Brush [18], Medium [17], AnimVR [35], Quill [31], MasterpieceVR [41] and Tvori [47]. These systems allow people to create virtual sketches in a free manner or provide an interface for animating drawings. However, the interfaces of these consumer tools are not user-friendly to non-experts and require a learning curve. HCI researchers have been working on various

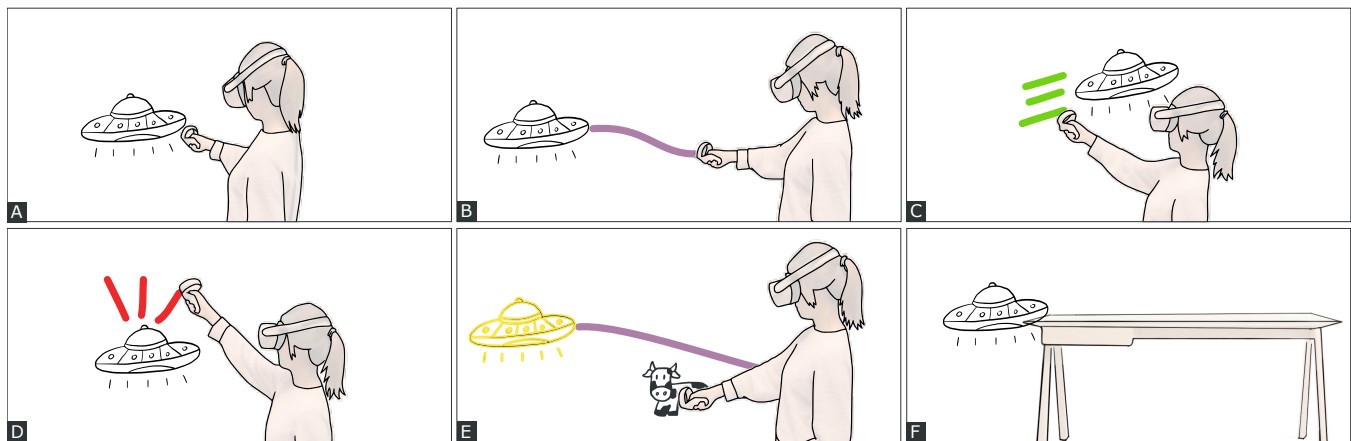

Figure 3: Interaction techniques. A) Drawing an element. B) Drawing a path. C) Drawing motion lines. D) Drawing sounds. E) Selecting and creating a dependency relationship. F) Interacting with physical surroundings.

system designs in order to have a better solution for a sketch-based interface. For example, HoloARt [1] investigates how people could paint in the air via finger gestures. VRSketchIn [8] uses interchangeable techniques of 2D and 3D sketching to create artifacts in VR. Other sketch-based 3D painting and modeling tools such as Cave-Painting [25], SweepCanvas [30], systems for model retrieval [11], Mobi3DSketch [27], and ChalkTalk [37] all provide users with a designed system to work in a virtual 3D scene. Immersive sketching tools including SymbiosisSketch [2] and PintAR [10] can create sketched elements that spatially integrate with the real world. RealitySketch [42] provides an AR interface to create embedding dynamic and responsive visualization within the physical environment. A bi-directional sketching interaction is introduced in Sketched Reality [19] that virtual objects and physical robots affect each other through AR sketches and tangible interfaces.

While these works try to address interactive sketching and design using different approaches, users need to manually specify the visual and audio effects at each step of the animation. The existing works do not take advantage of the semantic properties of sketched elements in the authoring process. Also, most of them do not support real-time interactions with the physical environment. In contrast, EnchantedBrush investigates a mixed-initiative interaction paradigm and uses the recognition of sketched elements to fill the unexplored area in these works. With our approach, users can create interactive animation in a real-world setting with expressive sound effects. Our contribution is a novel interaction approach in mixed reality using interactive motion and sound brushes for animating a storyboard.

## 3 ENCHANTEDBRUSH

The goal of EnchantedBrush is to allow people to visualize their ideas quickly and easily in an MR environment. To this end, we studied a set of grammar that is commonly used in storyboards, comics, illustrations, and icons. We found that motion and sound effects are two key components that make a scene dynamic, and creators and storytellers rely heavily on visual iconography to present those two effects as introduced by Scott McCloud in the book *Understanding Comics* [33]. Figure 2 displays three motion or sound examples. Therefore, we focus on the visual and sound design for animated storyboards. We propose a mixed-initiative interaction paradigm for motion and sound effects authoring. More precisely, our system automatically associates a default motion effect and a sound effect to each sketched object by leveraging its semantic nature. For example, the default motion effect for an airplane is flying around, and its default sound effect is an engine sound. Furthermore, to enhance the expressiveness of our system, we allow users to customize the motion and sound effects based on their own design idea.

The remainder of this section is organized as follows. We first introduce the design concepts of EnchantedBrush. Next, we demonstrate using EnchantedBrush for animation by a storytelling example.

### 3.1 Concepts

Based on *Understanding Comics* [33], we observed that storyboards are usually built upon the following concepts: *Drawing Elements*, *Drawing Motion*, *Drawing Sound*, *Selecting and Creating Relationship*. Accordingly, we design *Sketch* Brush, *Path* Brush and *Motion* Brush, *Sound* Brush, and *Selection* to achieve each concept, respectively.

#### 3.1.1 Drawing Elements

*Sketch* Brush allows users to draw freely in the air (Figure 3A). Once a sketched element is finished, the back-end sketch recognition model will take the sketch as input and send back the sketch recognition results to the user.

#### 3.1.2 Drawing Motion

*Path* Brush and *Motion* Brush are used to animate the sketches. There are two behavior modes in EnchantedBrush, automatic mode and customized mode. In the automatic mode, the system animates the sketch automatically based on sketch recognition and gives it a common behavior. This releases users from manually prescribing how an element should be animated. The common behavior is decided by a motion verb commonly associated with the given element. For instance, if the drawn element is a basketball, it will fall and bounce on the ground, while an airplane will fly around by default. *Path* brush is designed to use in the mode of customized behavior when the user wants their sketched elements to follow some specific trajectory (Figure 3B). When a customized path is provided, the system will switch to the mode of customized behavior and animate the element following the provided path. In either mode, the animation is started with motion lines drawn by *Motion* Brush (Figure 3C). The length of motion lines is parameterized and proportional to the element's movement speed. Longer motion lines cause the element to move faster so that users can control the element's speed and adapt to the unique need of their story.

#### 3.1.3 Drawing Sound

EnchantedBrush supports sound effects by incorporating lively audio. This enhances the expressiveness of storytelling and the sense of immersion. Upon the recognition of a sketched element, each element is automatically assigned three sound properties aligned with their semantic nature:

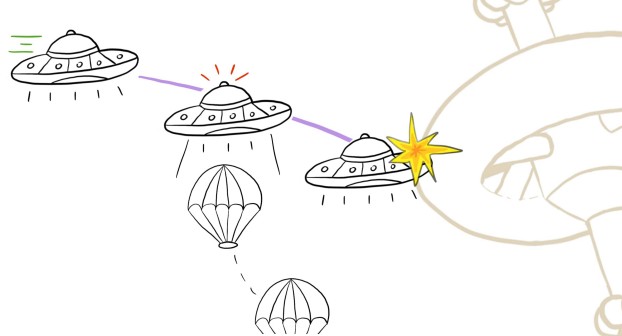

Figure 4: Example storyboard: aliens escape from a UFO before it collides with a space station (real-world objects replace beige elements).

1. **Self sound** is the unique sound commonly produced by an element. For example, the self-sound of an ambulance is a siren, while the self-sound of a dog is barking.

2. **Movement sound** refers to the sound made by an element while moving. For example, the movement sound of a car is an engine sound, and for a human, the movement sound is a walking sound.

3. **Collision sound** is the sound made when an element collides with another element. For example, the collision sound of a car is a crash sound, and it will be triggered when the car collides with either a physical object or a virtual element.

*Sound* brush aims to help users add the self sound of a drawn element (Figure 3D). Users can customize and specify when the self-sound should be played. For simplification, the sounds of movement and collision are played automatically once the element starts moving or a collision is detected.

### 3.1.4 Selecting and Creating Relationships

*Selection* enables users to select the element they want to edit and animate among multiple elements. It is implicitly activated based on the proximity of the brush to an element, and the selected element is highlighted with a yellow outline, as shown in Figure 3E. Once an element is selected, users can perform the animation on it using the designed brushes. In addition, to enhance the expressivity of the system, EnchantedBrush also supports inter-object animation by creating dependency relationships. For example, in Figure 3E, the UFO element is first selected, and a customized trajectory is provided for it. Users can then use the trajectory as a timeline and draw another element (i.e., the cow) along the line as if inserting a keyframe in the timeline. The cow is treated as a dependency of the UFO, and its appearance depends on the movement of the UFO. Moreover, as illustrated in Figure 3F, the sketched virtual element (i.e., the UFO) can interact with the physical surroundings (i.e., the table). Users can use physical objects for object models and backgrounds in their story which simplifies the story creation.

## 3.2 Composing Animation

In this section, we demonstrate how EnchantedBrush visualizes the story in Figure 4 using the interaction concepts introduced above. Figure 5 illustrates the design process.

We start with sketching a UFO element (Figure 5A) with *Sketch* Brush. Since we want the element to move in a specified behavior, we switch to *Path* Brush and draw a trajectory to make it collide with the shelf (the purple line in Figure 5B). The alien escapes by

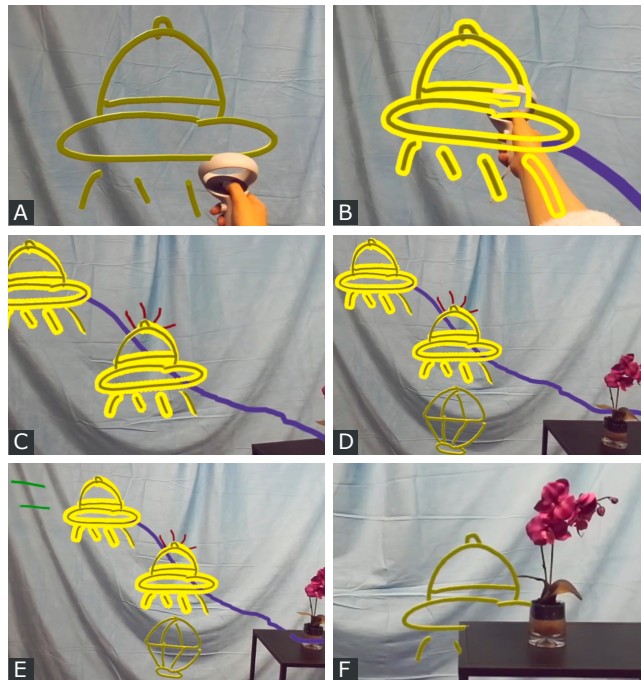

Figure 5: Steps to visualize a storyboard using EnchantedBrush: A) Sketch the main element of the story. B) Provide a customized trajectory (purple line) and select the animated element. C) Add the sound effect of an alarm. D) Sketch the dependency element (the parachute below the UFO). E) Animate the storyboard. F) Auto-play the movement sound and the collision sound.

parachute before the accident happens. The position of the appearance of the parachute depends on the movement of the UFO, so we compose a relationship between the UFO and the parachute by selecting the UFO (Figure 5B). When the escape happens, the UFO makes an alarm sound, and we can draw this by using *Sound* Brush (Figure 5C). We then switch back to *Sketch* Brush and draw the parachute element where the escape occurs (Figure 5D). Now the story is ready to be animated, so we switch to *Motion* Brush and animate the story using motion lines (Figure 5E). The movement sound is played automatically when the UFO starts moving, and the collision sound is played automatically when it crashes into the space station, i.e., the shelf (Figure 5F).

## 4 PROTOTYPE IMPLEMENTATION

We developed a prototype for the proposed concepts. In this section, we describe the implementation overview and detail the important components of our system including spatial mapping, sketch recognition, and object automatic behaviors.

## 4.1 System Overview and Setup

Our system requires two hardware components: Oculus Quest 2 as the Head-Mounted Display (HMD) and ZED Mini as the mixed-reality camera. The ZED camera is mounted on top of the Oculus HMD so that the virtual world resides in the real world. We use Oculus controllers as the input device.

Similar to SymbiosisSketch [2], we configure the setup of EnchantedBrush based on the bimanual design of painters in real life — painters use their dominant hand to hold the paintbrush, and the non-dominant hand to hold the palette. In our case, the controller held in the dominant hand acts as the main paintbrush while the other controller in the non-dominant hand acts as the palette which is a set of switchable brushes introduced in Section 3.1. Figure 6 shows

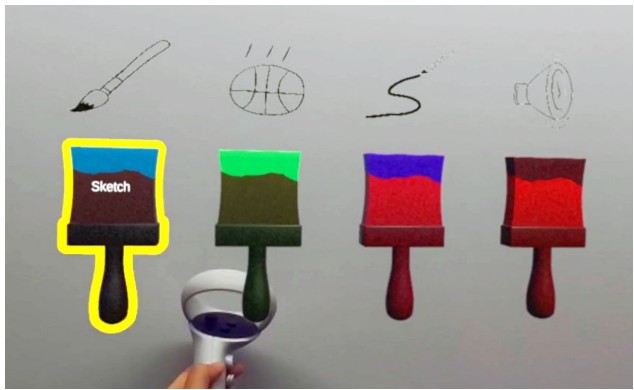

Figure 6: The menu of motion and sound brushes. Left to right: *Sketch* Brush, *Motion* Brush, *Path* Brush, and *Sound* Brush.

the interface design for this brush menu in our system. The user can use the controller (the main paintbrush) to select from *Sketch* Brush, *Motion* Brush, *Path* Brush, and *Sound* Brush in order to sketch elements, add motions, add customized paths, or add sounds. The system is implemented in the Unity engine by C#.

### 4.2 Spatial Mapping

We mark the location of physical objects in order to achieve interaction with the physical world. By using the built-in spatial mapping function of ZED Mini, we scan the real-world environment and model it in the 3D triangle mesh format. After the system stores the mesh of the real-world environment, we make it invisible to the users.

### 4.3 Sketch Recognition

In order to achieve the "enchantment" of EnchantedBrush, we implement automatic sketch recognition capabilities for our system through a sketch classification neural network containing three convolution layers and two dense fully connected layers. Due to the lack of publicly available dataset of multi-category 3D sketches, we train the neural network using eight categories of a 2D sketch dataset, Quick Draw Dataset from Google[1]. The eight categories include `parachute`, `cloud`, `basketball`, `cow`, `ambulance`, `police car`, `flying saucer`, and `airplane` based on our needs. We use 4,000 images for each category because of memory limitations. With this 2D network, we convert the 3D sketch into a 2D sketch before passing them to the sketch recognition network. To achieve this, we first project the 3D points onto a 2D best-fitting plane, then render the projected points into a normalized image. The accuracy of performance recognition is discussed in Section 6.4 in detail. The neural network returns two candidates of prediction according to its confidence in the accuracy of recognition. Figure 7 demonstrates this sketch recognition process. Once obtaining the recognition results from the neural network, we display the two results to the users and ask them to select the desired one.

### 4.4 Automatic Object Behaviors

If users do not provide a customized trajectory, the sketched element will be animated following its automatic behavior. A question raised here is: how do we decide what the automatic behavior of a given object should be? Before answering this question, we need to first define what "automatic behavior" is. Within the scope of EnchantedBrush, we define the automatic behavior of an object to be the common motion that is widely associated with the object. For

---

[1]The Quick Draw Dataset: `https://github.com/googlecreativelab/quickdraw-dataset`

example, the automatic behavior of a ball would be bouncing, while for a car, the automatic behavior would be running forward. To enhance the scalability, we experimented with the powerful language model Generative Pre-trained Transformer 3 (GPT-3) developed by OpenAI [5] to generate a motion verb for a random object and use the verb as its automatic behavior. Once we obtain a common verb associated with the given object, we transform the verb into an animation effect. The translation from a verb to a motion path is implemented as changing the position vector of the object in Unity. Take an airplane as an example, we first get the verb *fly* from GPT-3, then we transform it into a flying behavior: we define the flying motion as a circular path and the rotation around the y-axis in degrees per unit time.

Upon the recognition of a sketch, the corresponding sound effects are automatically assigned to the sketched element. We use local pre-downloaded audio resources in the prototype. In detail, for each supported object category, we downloaded self, movement, and collision sounds. When the system is notified of the object identity, it retrieves the sound effect locally and assigns it to the sketched element.

## 5 EVALUATION

We conducted an exploratory user study with both general users and professional artists. The first goal was to evaluate our approach and interaction techniques; the second was to identify limitations and potential applications of our system.

### 5.1 Participants

We recruited 12 participants (eight females) aged 23 to 34 to evaluate our system. Three participants are professional artists (P7, P11, and P12). Participant P7 has eight years of professional experience in illustration and sketching, P11 has 11 years of experience in architecture and urban design, and P12 has six years of experience in animation and design. The other participants have little to good experience with sketching and storytelling and minor to moderate experience with animation.

### 5.2 Tasks

Participants were given four tasks to work on using EnchantedBrush. The first three tasks were simple animated scenes that aim to familiarize participants with the system's main features. The last task was to recreate a short story using EnchantedBrush once participants felt confident about using the system. The four tasks are the following:
**Task 1.** (Figure 8A): participants were required to draw a bouncing basketball on the top of a physical table. The basketball was

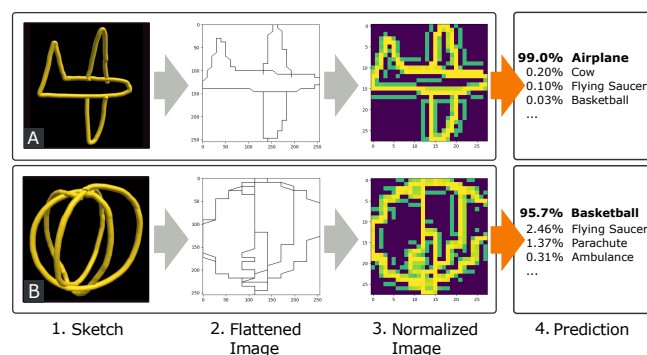

Figure 7: Sketch recognition process: the 3D strokes are first projected onto a 2D best-fitting plane to get a flattened image. Flattened images are then converted to normalized bitmaps. The recognition model offers a list of predictions with confidence based on the bitmaps. Sketch A is an airplane, and Sketch B is a basketball.

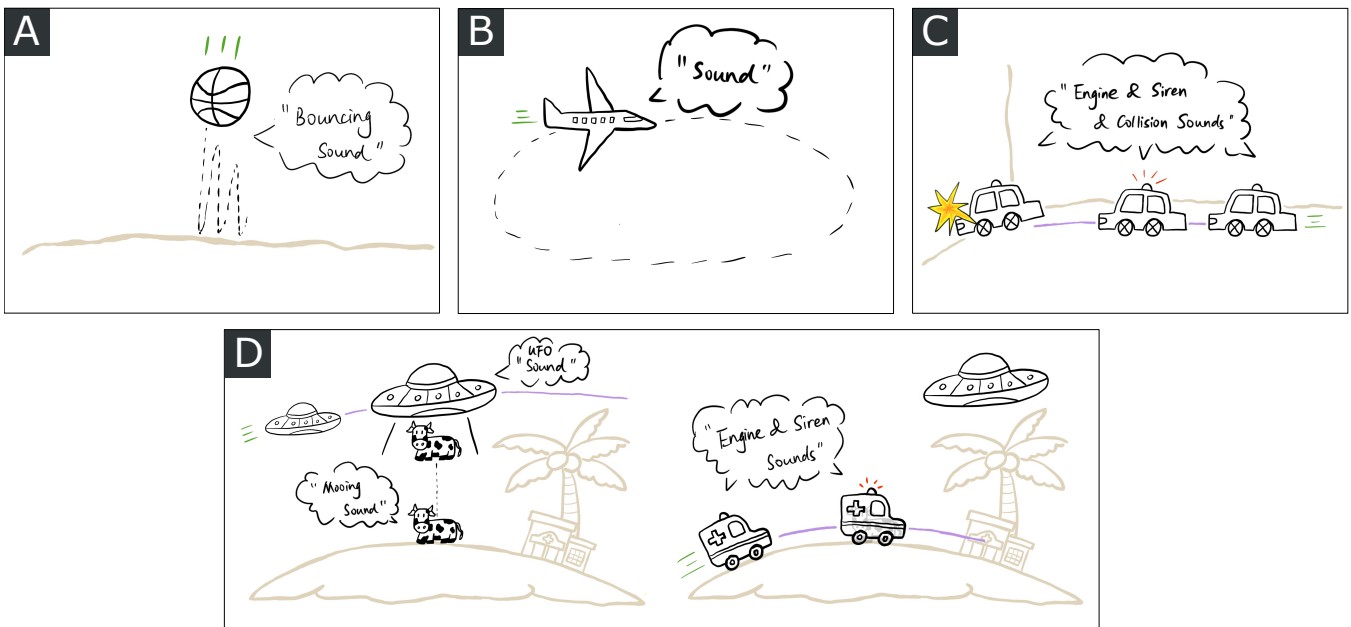

Figure 8: Storyboards used in the evaluation sessions (real-world objects replace beige elements): A) A ball bounces on the table and makes a bouncing sound. B) An airplane flies around with some engine sound. C) A police car moves forward and collides with a wall. An engine, a siren, and a collision sound are playing in the process. D) A poor cow is abandoned on an island by a UFO. An ambulance then takes the injured cow to the hospital. The sounds of the cow, the UFO, and the ambulance are playing in the process.

animated using its automatic behavior. Participants would hear the auto-played bouncing sound. *Sketch* Brush and *Motion* Brush were used in this task.

**Task 2.** (Figure 8B): participants were required to draw an airplane that flies around. This task used automatic behavior to animate the airplane element. Participants would hear the jet engine sound while the airplane was flying. Similar to Task 1, *Sketch* Brush and *Motion* Brush were used in this task.

**Task 3.** (Figure 8C): participants were asked to create a car crash story. A physical curtain replaced the wall that the car collides with. *Sketch* Brush, *Motion* Brush, *Path* Brush, and *Sound* Brush were all used in this task.

**Task 4.** (Figure 8D): participants were asked to create a more complicated short story. *One day, a UFO flew over an island, and as it passed over the island, it abandoned a poor cow that had been kidnapped. Fortunately, there was an animal hospital on this island. An ambulance arrived in time for the injured cow and took it to the hospital.* In addition to the four brushes, *Sketch* Brush, *Motion* Brush, *Path* Brush, and *Sound* Brush, *Selection* was also used to select the UFO so that the user could sketch the cow as a dependency of the UFO. Existing physical objects, such as a table and flowers, were used as the island and the animal hospital, respectively. Thus, participants could focus on the main story plot, saving their time making models of islands or hospitals.

### 5.3 Procedure

The study was conducted in our lab. Pieces of furniture and objects, including a table, a shelf, and a flower, were used to set up the study space so that participants could use them while creating a storyboard. The example storyboard for each task (Figure 8A-D) was displayed on a separate monitor, and participants could refer to them whenever needed.

Our evaluation session consisted of three steps. First, participants filled out a background questionnaire. The facilitator then introduced the user interface and core concepts and demonstrated the functionalities of each brush by going through Task 1 to Task

3. Participants then took time to familiarize themselves with the interface and interaction methods. In the second step, participants were asked to perform the four tasks independently. The facilitator provided light guidance if the participant had trouble using the tool. There was no time limit on task completion, so participants could pay full attention to the tool's usability without pressure on completing a task on time. Finally, participants were asked to fill out a usability questionnaire and participate in an interview discussion to collect more in-depth insights about our system. Sessions lasted approximately 60 minutes, and participants were compensated with 20 CAD.

## 6 RESULTS AND DISCUSSION

Our evaluation suggests that participants enjoyed using Enchanted-Brush to create storyboards and animate their ideas. Participants appreciated the simplicity of the interface and the easiness of animation authoring. We also studied the potential applications of EnchantedBrush according to the discussions with the participants. Figure 9 shows two examples of participants during our user study, and Figure 10 shows four sample results produced by our participants.

### 6.1 Quantitative Metrics

We summarized the user preference results from the questionnaire in Table 1. We took the System Usability Scale as a reference and tailor-made our questionnaire specifically for our focus of attention, i.e., the user experience with the proposed features and storytelling. We used a set of 1-10 Likert scale questions for the measurement (1 = strongly disagree to 10 = strongly agree). Overall, participants found it easy to use EnchantedBrush to create a story (Q1). They were satisfied with the interaction techniques, including triggering a sound effect (Q3), making an object follow a given path (Q4), and using motion lines to animate an object (Q5). At the same time, we found the easiness level of sketching an object was lower than the other features (Q2). Through discussions with the participants, they suggested they were not used to the precision and control over

Table 1: Results for the questionnaires of user preferences (Median and Interquartile Range).

| Statements | | Median (IQR) |
|---|---|---|
| I felt... | | |
| 1. | it was easy to create a story. | 9 (3) |
| 2. | it was easy to sketch an object. | 7.5 (2.25) |
| 3. | it was easy to trigger a sound. | 10 (1.25) |
| 4. | it was easy to create a trajectory. | 10 (1) |
| 5. | it was easy to start a story using motion lines. | 9 (1.25) |
| 6. | the interface was simple and easy to use. | 10 (1.25) |
| 7. | the real-world environment supported the story. | 9 (1.25) |
| 8. | the EnchantedBrush made storytelling easy. | 9 (1) |
| 9. | the EnchantedBrush made animation authoring easy. | 9 (2.25) |

Strongly Disagree | 1 | 2 | 3 | 4 | 5 | 6 | 7 | 8 | 9 | 10 | Strongly Agree

sketch strokes when drawing in 3D space. This finding echoes the discussions made by previous works [2, 3, 24] that it is a general challenge for humans to draw accurately in the air due to the inability of ergonomics. In terms of the user interface, participants all agreed on the simplicity of the interface and suggested that it was easy and straightforward to use (Q6). Participants felt that interacting with the real-world environment made it easy to tell a story (Q7). All participants were confident that EnchantedBrush made storytelling and animation authoring easier (Q8, Q9).

The questionnaire results showed that EnchantedBrush provides users with an easy sketch-based interface to create storyboards and communicate their ideas. Sound components, motion lines, customized trajectories, and interactive mixed-reality environments were appreciated for their power and effectiveness.

## 6.2 Qualitative Feedback

We conducted guided interviews and open-ended discussions with the participants about their user experience to collect qualitative feedback and insights. We summarize the common points they mentioned and present the highlights as follows.

Overall, participants showed great excitement about our tool and how they could create stories in the real world. For example, P5, an amateur artist who studied animation for two years and has experience sketching for about ten years, appreciated how they could use physical objects for animation which freed them from making models of objects. P12, a professional animator, also liked the mixed-reality environment and pointed out that seeing the physical world contributed to the magical senses of the tool.

*P5: "Normally, when I animate things, it's all digital. If I want a wall for the car to crash into, I have to make the wall. I have to put it in the right spot and figure out how to make it part of the story. But since the wall is already there, I think using it as part of the story is clever. Same with the table. I felt like if you're animating, you need to figure out the contact and the rigid body if you're making a ball down, so it's cool that it already knew how to use the table."*

Professional artists (P7, P11, P12) commented that Enchanted-Brush provided a simple and interactive interface and believed that EnchantedBrush made dynamic planning, idea presentation in front of a team, and communication more effective.

*P7 (pro): "I had a lot of fun using it [...] It'll be so easy to draw a thing that's interactive, and your whole team can see it and brainstorm a lot easier [...] You are being able to do a dynamic plan and brainstorming with your whole story-boarding team. I think it'll make the connection between the authors (people making up*

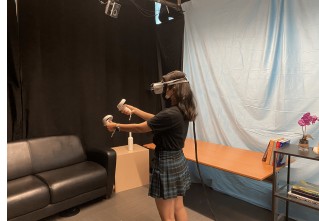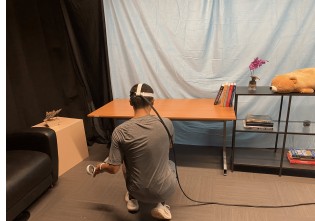

Figure 9: Participants working on the tasks.

*the story) and people making it come to reality... And it'll be easier, quicker, more organic."*

*P12 (pro): "I think that it was quite amusing and inspiring to see things that you draw come up to life [...] You can test certain scenarios that you're trying to build for [such as] animation, like just straight out very quickly, and see how it's actually like, without any dialogue or any other function. Just by the movement, you can tell a story."*

They also expressed strong interest in having such a tool in their career work.

*P11 (pro): "It was very easy and fun to use. I sort of wish I had [this when] I worked in After Effects as an architectural designer. It would be amazing if I had something similar to this for the purpose of presenting initial animation ideas [...] Having a tool like this would really help the concept formation, or the storyboard formation phase [and] will save everyone a lot of time on work. [An example is like] I want you to show the buildings in relationship to the public space [...] I would spend a week or two weeks working with other people who were doing the modeling, trying to assemble the thing, but then when we showed it to our supervisor, it was not what he wanted, so we had to do it all over again [...] What to show, what elements to show, and what components to show at what angle is really important, so I see work like this is a really good future in saving people's lives about not doing repetitive work."*

The qualitative feedback we collected indicated that participants found EnchantedBrush useful. They enjoyed the core concepts of our system. Also, we did not find any issues with the real-world mapping of physical objects that affected user experience.

## 6.3 Potential Applications

All the participants believed in the potential benefits of EnchantedBrush for different applications and use scenarios. Animation

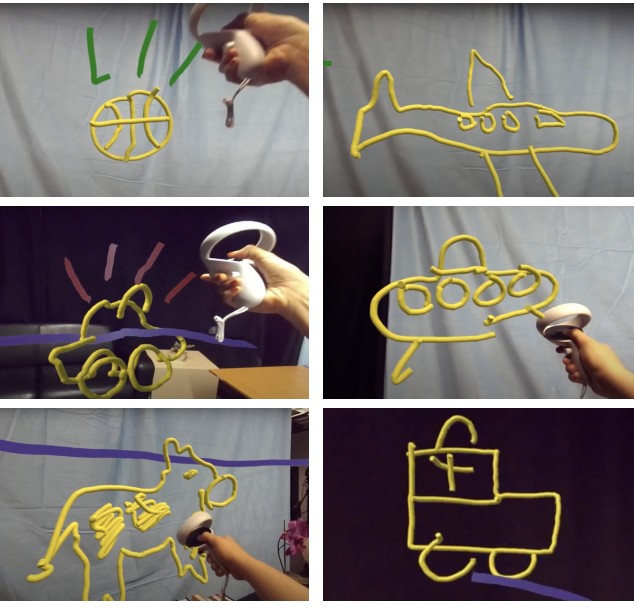

Figure 10: Sample results created by participants in the task sessions. Left to right and top to bottom: a bouncing basketball (Task 1), an airplane (Task 2), a car with a siren sound (Task 3), and a UFO, an abandoned cow, and an ambulance (Task 4).

industries are one of the fields that our participants mentioned. *"In the animation industry, you need to actually draw every frame. But in this case, even if you don't draw every frame, it sort of copies it (i.e. frames) over and has a moving effect, so I think I can see some potential."* (P1). Seven (out of 12) participants stated that such a tool would be appealing to children, so it could be useful in educational fields such as student engagement, concept demonstration, and class teaching. P3 pointed out that the power of quick storytelling of EnchantedBrush could help teachers reproduce a story scenario for children, so children could not only comprehend the story from oral descriptions but also experience the story themselves to enhance their understanding and creativity from visualizations. Another point suggested by P3 is that children could learn about different physical properties (such as gravity, materials, and sounds) of an object using EnchantedBrush based on the interaction and visual effects between virtual objects and real-world environments. Besides, P11 commented that EnchantedBrush would also be useful in social media and video making. *"When people go on Instagram live or TikTok live, and they tell a story, and then people get bored about just listening to them talking about the story and showing their face without anything going on. I see this thing as like when you listen to profs giving lectures, sometimes you get bored. That's why they draw stuff on the blackboard and why they used animation slides [...] But if they could use your tool and actually draw out the story, I'm pretty sure you will attract so many more listeners."* (P11).

### 6.4 Sketch Recognition Performance

The sketch recognition model was trained on 2D drawings, so the accuracy of recognizing 3D sketches is not perfect. We achieved on average 98.83% recognition accuracy on the 1,000 testing images from Quick Draw Dataset and 68.0% on 3D sketch recognition in our study. We also analyzed the performance based on each task: Task 1 has an accuracy rate of 100%, Task 2 and Task 3 are 58.3%, and Task 4 gets an accuracy rate of 63.89%. We found that the model worked perfectly on recognizing a basketball, but the accuracy of other objects varies significantly from person to person due to various sketching skills, styles, and the deformation

caused by conversion from 3D sketch to 2D sketch. Since sketch recognition itself is not one of our main focuses, and to allow users to evaluate our interaction approach in a better way, the false results were corrected manually during the sessions.

## 7 LIMITATIONS AND FUTURE WORK

Although we demonstrated that EnchantedBrush enables users to create storyboards and animate ideas easily, there are a few limitations along with opportunities for future work and research.

In our current prototype, the sketch recognition neural network is trained by a 2D sketch dataset. This limits the sketch recognition accuracy of our system. With the emergence of large-scale datasets of multi-category 3D sketches in the future, we believe sketch recognition accuracy could be improved. Meanwhile, the current sketched elements are relatively flat, as sketching 3D-shaped objects in mid-air is generally challenging for users. The MR design space could be augmented so users could sketch more 3D-shaped elements more easily. Besides, the number of object categories that our current prototype supports was small since we focused on evaluating our approach and validating the concepts and the implemented features. Expanding the scale of the object categories could support users' creativity and further leverage the interaction techniques of EnchantedBrush. Lastly, the retrieval of sound effects could also be expanded to use a text-based audio retrieval method from the internet rather than using pre-downloaded audio files.

In our current design, we focused on assisting users to easily control the animation of the sketched elements. It would be also interesting to support the deformation of sketched elements according to their unique physical properties or materials in reaction to contacts and collisions. One possible direction is to explore how to assist users to create this casual physics-based deformation in MR. This could allow users to achieve more realistic scenes.

Another research direction is to further automate our proposed interaction techniques by automatically interpolating users' intended animation effects. In our approach, we designed various brushes for motion and audio effects. Future work could be to leverage human knowledge and more visual languages in comics design and storytelling to help machines understand the design intent of users. With more visual languages and text-based audio retrieval methods mentioned above, users would be free to create more diverse stories and objects such as talking animals. This could also save users from the explicit use of brushes and thus lead to a more powerful and free-form tool for animation authoring.

## 8 CONCLUSION

In this paper, we presented EnchantedBrush, a novel mixed-reality sketching interface for animating storyboards in real-world environments with automatic sound effects. We proposed a mixed-initiative interaction paradigm for motion and sound effects based on the semantic nature of sketched elements, which fills the gap in the existing works. Our proposed interaction paradigm allows users to quickly create visualizations and storyboards without spending intensive time on manual effect specifications. EnchantedBrush also supports storyboards to interact with the physical surroundings which simplifies the creation process. Finally, our user study suggests that our approach can be an easy and effective tool for storytelling sketching with a variety of potential applications.

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
