# OpenReview forum: "EnchantedBrush: Animating in Mixed Reality for Storytelling and Communication"
_graphicsinterface.org/Graphics_Interface/2023/Conference_SD — GI 2023 - second deadline_

### Official Review · Reviewer_24c1 · 2023-04-12
**Review of EnchantedBrush**

**Rating:** 7
**Confidence:** 5

**Review:**

This paper presents EnchantedBrush, a mixed reality system for creating animated storyboards using motion and sound effects in the physical world.  The paper presents the details of the EnchantedBrush interface as well as a user study with 12 participants that included professional artists.  The results of the study suggests that the system makes it easy to create simple storyboards that utilizes the physical environment as well as areas for future work and improvement.

Overall, this paper presents interesting work in an area that has received considerable attention in recent years.  The related work is sound, and the paper is generally well written and easy to follow. The user study was well conceived and carried out and provides some interesting insights into the features and functionality of EnchantedBrush.

There are several issues with the paper that should be addressed as there is missing details in a number of areas.  The first issue is with the description of the sketch recognition.  The sketch recognizer uses a neural network, but it is unclear what kind of neural network.  In addition, how much training data was used? How many different objects can the recognizer classify? It appears from the paper that there only about 5 or 6 different objects that can be recognized. If so, it seems that the recognition accuracy should be much higher than it is. The authors should present more details in the paper to answer these questions.  Another question is with sketches users draw for creating sounds and animations.  At first glance it seems that users can sketch these out and the system recognizes them.  However, when watching the video, it appears that there is a menu system that helps to break these operations into chunks.  This menu system is not discussed in the paper or how it is used to delineate different motion and sound gestures. This information should be in the paper as well.

The user study presents some nice information on the usability of the system, but it is unclear why these usability scores where so high, given the poor recognition rates.  It would seem that this would potentially reduce usability of the system.  In addition, task 4 is clearly the most interesting task that users had to do in the study.  It would be nice if the authors put an image or two of one of the study participants' T4 drawings.

In spite of these issues, I believe the paper can be revised to include these missing details and should be considered above the bar for GI.

---

### Official Review · Reviewer_szzR · 2023-04-20
**Interesting paper despite some weaknesses**

**Rating:** 6
**Confidence:** 4

**Review:**

This paper presents a platform to perform sketching in augmented reality. The platform allows the creation of animated storyboards that can interact with the physical surrounding. The approach allows to combine automatic object behaviour definition and manual modifications of these behaviours. The paper presents the concept, implementation and evaluation of this approach.


The paper is well-written and easy to follow. The initial motivation is clearly presented and the idea of combining immersive sketching with the physical surrounding is interesting and sound. The paper covers the necessary steps of exploring such a concept, as it details the concept's key aspects, the implementation and the evaluation of the system.

I liked how the authors based the concepts on the well-known work by Scott Mcloud.

There are some limitations in the development of the automatic mode, mainly that it is trained on a 2D sketch dataset, but this is acknowledged by authors and does not change the system concept. I think the other mentioned limitations (number of object categories) are reasonable for a proof of concept system as Enchanted Brush.

One comment regarding novelty is that the paper does not clarify which features are novel per se (a table could help). While I understand that the automatic mode and the interaction with the physical environment are the key novel elements, authors could point to previous work for each feature (e.g. drawing elements).

The evaluation is relatively informal. Participants had to perform four rather simple tasks. I would have liked more motivations about why authors used a homemade usability questionnaire instead of one of the well-defined already existing questionnaires. The questions seem somewhat biased towards having a positive opinion of the system (i.e. they all use a positive term such as "easy"). This is a major weakness of the evaluation and the paper.

I think authors need to tone down the claims made following the results of this questionnaire, such as "The questionnaire results demonstrated the usability of our tool".

it is not said how did authors analyse the qualitative feedback.

To sum up, the paper presents a nice and interesting tool. The major weakness of the paper is the tool's evaluation. The paper would require minor rewriting to tone down some overclaims and clarify that this questionnaire is a weak aspect of the study.

---

### Official Review · Reviewer_ET9a · 2023-04-23
**Nice system and decent evaluation, but the contribution needs to clearer**

**Rating:** 6
**Confidence:** 4

**Review:**

Summary of recommendation: the paper contains a reasonable collection of parts that add up to a novel interactive system, although the authors need to do more to specify exactly what is novel compared to previous work, and need to do more to discuss the limitations of the system. Overall, the paper could be above the bar, but further work is needed to improve the framing and presentation.

The authors present a system for prototyping mixed-reality animations, using sketching, sketch recognition, and automated mechanisms for identifying real-world terrain and assigning motion paths and sound effects. There are many previous systems that exist in this space, and the paper does a reasonable job of surveying these - however, more needs to be done to specify exactly what the limits of previous systems are. The authors imply that the main restriction of previous work is in the automated attachment of sound and motion paths, but if this is the main area of contribution for the submission, the authors do not expand on these capabilities sufficiently to be able to judge the improvement on the state of the art.

Assuming that attaching motion and sound to sketched objects is the primary contribution, there are several issues that arise from the paper. For example, there is no clear assessment of the range of objects that can be handled by the system. The paper states "for each supported object category, we downloaded self, movement, and collision sounds" and later, "the number of object categories that our current prototype supports was small". The authors need to clearly specify what range of objects could be handled by the system: a range of questions arise about how well this critical part of the system actually worked - how often does the system get the path / audio wrong for the recognized object? how often is the path / audio different from what the user intended? what happens if the user draws a shape for which there is no audio / path? All of these issues would greatly reduce the "enchantment factor" of the system and so should be addressed in more detail. (As an aside, what would the system do with anthropomorphized objects? Mickey Mouse does not act like or sound like a mouse - how would the system handle this? What if the cow in Task 4 was a talking cow?). Concerns about the capabilities of the automated sound / path mechanism are highly relevant if this is in fact the main area of contribution for the paper.

There are also other concerns that need to be addressed. The authors state that real-world mapping is done by an off-the-shelf piece of technology, but do not indicate how well this works and whether problems arise for the user's experience (e.g., if the user changes location, does the perspective shift mean that all of the motion paths are now in mid-air?).

Overall, this seems like it is an interesting system that could be an advance over current capabilities - but without a better understanding of exactly what the authors are claiming as a contribution, and how well the system works in these areas, it is difficult to judge the value of the paper to the HCI research community.